# Protocol for validation of the 4AT, a rapid screening tool for delirium: a multicentre prospective diagnostic test accuracy study

Susan D Shenkin,[1] Christopher Fox,[2] Mary Godfrey,[3] Najma Siddiqi,[4] Steve Goodacre,[5] John Young,[6] Atul Anand,[7] Alasdair Gray,[8] Joel Smith,[9] Tracy Ryan,[10] Janet Hanley,[11] Allan MacRaild,[12] Jill Steven,[12] Polly L Black,[12] Julia Boyd,[13] Christopher J Weir,[13,14] Alasdair MJ MacLullich[15]

► Prepublication history and additional material is available. To view please visit the journal (http://dx.doi.org/ 10.1136/bmjopen-2016-015572).

For numbered affiliations see end of article.

**Correspondence to**
Professor Alasdair MJ MacLullich;
a.maclullich@ed.ac.uk

## ABSTRACT

**Introduction** Delirium is a severe neuropsychiatric syndrome of rapid onset, commonly precipitated by acute illness. It is common in older people in the emergency department (ED) and acute hospital, but greatly under-recognised in these and other settings. Delirium and other forms of cognitive impairment, particularly dementia, commonly coexist. There is a need for a rapid delirium screening tool that can be administered by a range of professional-level healthcare staff to patients with sensory or functional impairments in a busy clinical environment, which also incorporates general cognitive assessment. We developed the 4 'A's Test (4AT) for this purpose. This study's primary objective is to validate the 4AT against a reference standard. Secondary objectives include (1) comparing the 4AT with another widely used test (the Confusion Assessment Method (CAM)); (2) determining if the 4AT is sensitive to general cognitive impairment; (3) assessing if 4AT scores predict outcomes, including (4) a health economic analysis.

**Methods and analysis** 900 patients aged 70 or over in EDs or acute general medical wards will be recruited in three sites (Edinburgh, Bradford and Sheffield) over 18 months. Each patient will undergo a reference standard delirium assessment and will be randomised to assessment with either the 4AT or the CAM. At 12 weeks, outcomes (length of stay, institutionalisation and mortality) and resource utilisation will be collected by a questionnaire and via the electronic patient record.

**Ethics and dissemination** Ethical approval was granted in Scotland and England. The study involves administering tests commonly used in clinical practice. The main ethical issues are the essential recruitment of people without capacity. Dissemination is planned via publication in high impact journals, presentation at conferences, social media and the website www.the4AT.com.

**Trial registration number** ISRCTN53388093; Pre-results.

### Strengths and limitations of this study

► The study protocol involved seeking a representative sample of older acute medical patients in the emergency department and acute medical wards. A detailed, structured reference standard with explicit and reproducible methods is used to assess the features of delirium and reach a diagnosis.

► Two different rating scales, the 4 'A's Test (4AT) and the Confusion Assessment Method are being evaluated in similar groups of patients.

► Reference standard and index assessments were performed blinded to each other.

► A limitation of the study is that participants or legal proxies were required to give consent and thus the sample was selected.

acute deterioration in attention and other mental functions. The diagnostic criteria are, in summary: a disturbance of consciousness (ie, reduced ability to focus, sustain or shift attention) and a change in cognition. The mental status deterioration develops over short periods of time (usually hours to days) and it tends to fluctuate.[1,2] Delirium is commonly precipitated by acute illness, trauma or the side effects of medications. The presence of a 'medical condition' is part of the Diagnostic and Statistical Manual for Mental Disorders, fourth and fifth Edition (DSM-IV, DSM-5) criteria. Delirium is extremely common: it affects at least 15% of patients in acute hospitals and is more common in older people.[3–5] It is independently associated with many poor outcomes.[6–10] Delirium is also a marker of current dementia[6,11] and is associated with acceleration of existing dementia.[12] In older patients without dementia, an episode of delirium strongly predicts future dementia risk.[7,13] The economic burden of

## INTRODUCTION
### Background

Delirium is a severe and distressing neuropsychiatric syndrome which is characterised by

delirium derived from 2008 US data estimates the 1-year healthcare costs to be \$38–\$152 billion,[13] but there are limited recent data on the costs associated with delirium.

Detection of delirium is essential because it indicates acute systemic or central nervous system illness, physiological disturbance and drug intoxication or withdrawal. Failure to detect delirium in the acute setting is associated with worse outcomes.[14] Specific management of delirium is of obvious and immediate benefit to patients in many clinical situations, for example, in reversing opioid toxicity, treatment of peripheral infections which have presented with delirium, alleviating distress caused by delusions and hallucinations[15] and in prompting more thorough assessment of symptoms.[16]

More broadly, detecting cognitive impairment in general (delirium, dementia, depression, learning disability, etc) is a prerequisite for high-quality care because of the multiple immediate implications of cognitive impairment for patients and staff, including: ensuring adequate communication with patients and their families, doing careful assessment of capacity to provide consent for clinical procedures, avoiding giving treatments contrary to the law because of lack of consent, alleviating distress more readily, avoiding unnecessary bed transfers and prompting delirium prevention, including a detailed drugs review. Detection of dementia has recently been highlighted in the Dementia Commissioning for Quality and Innovation framework in operation in NHS England.[17]

In general medical and emergency department (ED) settings, delirium is grossly underdetected: at least two-thirds of cases are missed.[5 18 19] It is unclear why detection rates are so low. Evidence from surveys and workshops has raised several possibilities, including general ignorance about delirium, lack of awareness of its importance, uncertainty about discriminating delirium from dementia and lack of time for assessment in the acute setting.[20–24] The lack of a very rapid, simple and validated screening tool is a major factor in the underdetection of delirium.

Many delirium assessment instruments have been developed that operationalise the standard diagnostic criteria for delirium, but these have largely remained research tools. The most commonly advocated screening tool for use in routine clinical care, the short Confusion Assessment Method (CAM),[25] has satisfactory sensitivity and specificity in trained hands though takes around 10 min to complete because it requires a cognitive assessment like the Modified Mini-Cog[26 27] to be done first. The CAM also requires the rater to make subjective judgements of mental status. Subjective judgements are less reliable, often more time-consuming and more difficult for staff (particularly non-specialists) than simple objective measures with clearly defined cut points.[27]

The problem of some patients being 'untestable' is likely to be another important factor in delirium underdetection: many patients in acute settings are too unwell, sleepy or agitated to undergo cognitive testing or even

---

**Box 1    Requirements for a screening tool for delirium for use in the acute hospital environment**

- ► Short (less than 2 min)
- ► Easy to learn
- ► Easy to administer and score
- ► Can be used by professional-level healthcare staff from a variety of disciplines
- ► Allows scoring of patients who are too drowsy or agitated to undergo cognitive testing or clinical interview
- ► Takes account of informant history
- ► Can be administered through written questions to people with severe hearing impairment
- ► Can be administered to patients with visual impairments
- ► Does not require subjective judgements based on interview
- ► Combines delirium screening with general cognitive screening
- ► Does not need a quiet environment for administration
- ► Does not require physical responses such as drawing figures or clocks

---

interview.[28–31] Most screening tools do not make explicit how these patients should be classified. The result is that mental status assessments are often simply left uncompleted in most 'untestable' patients, and no diagnosis, and often no specific treatment, is applied. This lack of diagnosis can be harmful.[14]

Finally, given the time pressures in acute settings, it is challenging to implement a separate delirium screening instrument in addition to any existing general cognitive screening instruments. The lack of a combined instrument allowing screening for both general cognitive impairment and delirium may therefore contribute to the lack of specific delirium detection. Early diagnosis of delirium using evidence-based diagnostic tools offers a means for improved outcomes and more efficient resource allocation decisions.

### Rationale for the study

Given the multiple constraints of the acute environment, the range of staff that might be expected to screen for delirium, the common coexistence of delirium and dementia and the heterogeneity of patients, we determined the requirements for a screening tool (box 1).

There are multiple instruments for delirium screening, diagnosis, severity assessment and monitoring.[32–35] Before deciding to design a new screening tool, we therefore examined each of the available tools against the above criteria, focusing on screening tools such as the CAM. We also searched the literature systematically, including conference proceedings, books and book chapters, for any newly published tools as well as to examine the study data for each tool. Most scales were excluded on grounds of duration alone. The remaining scales lacked features such as general cognitive screening and other important features. We thus found that, in late 2010, no existing tool fulfilled the above requirements, and because of this we decided to design a new test. This conclusion was supported by

the National Institute for Health and Care Excellence (NICE) Guidelines on Delirium[6] which emphasised the need for research on a screening tool for delirium suitable for routine use.

The subsequent design process involved scrutiny of each of the nearly 30 published delirium assessment tools, evaluating the performance of each, including subtests, in published studies and, in most cases, through direct clinical or research experience of their use. Because we had decided to incorporate general cognitive screening into the new instrument, to avoid the need to have separate instruments for cognitive screening and delirium screening, we also reviewed the broader literature on brief tests for general cognitive impairment (including dementia). In the context of designing a screening tool for the acute hospital, it is important to note that delirium generally causes cognitive impairment detectable on the kinds of tests used for dementia screening.[36 37] Therefore, abnormal test results may indicate delirium and/or dementia (as well as other causes of cognitive impairment, such as learning disability).

It is clinically essential to know if any such impairment is acute, that is, delirium, but also important to identify underlying general (acute or chronic) cognitive impairment. A tool designed exclusively to detect cognitive impairment will not lead to delirium detection without another step, and a tool designed only to detect delirium may miss general cognitive impairment. In this light, we decided that the 4 'A's Test (4AT) should include cognitive screening sensitive to general cognitive impairment, but also including items on altered level of alertness and change in mental status, both of which are strong indicators of delirium.

The first version of the 4AT was drafted and tested informally by colleagues, changes were made based on feedback and updated versions were tested again. After several iterations involving 20 doctors and nurses of varying levels of experience, the final version was produced. An initial audit in 30 inpatients comparing clinical use of 4AT with independent reference standard DSM-IV assessment found 100% sensitivity (CI 69% to 100%) and 90% specificity (CI 68% to 99%). A subsequent validation study in Italy involving 234 consecutively recruited older hospitalised patients found that the 4AT had a sensitivity of 89.7% and a specificity of 84.1% for delirium.[38] The area under the receiver operating characteristic (ROC) curves for delirium diagnosis was 0.93. Since the 4AT was launched, locally and through the www.the4AT.com website, it has been adopted in clinical units in several centres in the UK and internationally.

Thus, in 2014, there was encouraging evidence that the 4AT has value as a tool for delirium detection in routine practice. This evidence came from several sources: one published study, audits in several sites, informal feedback, adoption in clinical practice by several clinical units globally and a recent web-based survey focused specifically on 4AT provided evidence supporting its use. Since this study was designed, other validation studies have been published, with favourable results; however, these included specific clinical populations (eg, stroke[39]), languages (Thai[40]) had relatively small numbers[41] or validated assessments against clinical assessment rather than research reference standard assessment.[42] Therefore, a formal, large validation study is necessary to provide definitive evidence of the diagnostic accuracy of the 4AT.

Comparison with the CAM is also of value, because the CAM is in use in some clinical units and thus information on how the 4AT performs in relation to the CAM will help clinicians decide which tool is suitable for their particular context. Further information on how the 4AT performs as a cognitive screening tool, its ability to predict outcomes and how each item of the 4AT contributes to its diagnostic accuracy will also provide important guidance to clinicians. Finally, understanding the economic costs and benefits of using the 4AT and the CAM will help providers in service pathway decisions.

### Study objectives

The *primary objective* of the study is to determine the diagnostic accuracy of the 4AT for delirium detection versus the reference standard of a DSM-IV diagnosis.

The *secondary objectives* are:

a. to compare performance of the 4AT and the CAM;
b. to determine if the 4AT is an adequately sensitive tool for detecting general cognitive impairment as judged against a documented history of dementia and/or the Informant Questionnaire for Cognitive Decline in the Elderly (IQCODE);
c. to determine if 4AT scores predict important outcomes such as length of stay, institutionalisation and mortality up to 12 weeks;
d. to determine the performance of individual items of the 4AT, for example, how accurate is altered level of alertness alone as a predictor of delirium diagnosis?
e. to assess the 4AT total score as a measure of delirium severity;
f. to estimate the delivery costs of the 4AT and CAM as a function of their diagnostic performance up to 12 weeks as well as modelling longer term resource consequences.

## METHODS AND ANALYSIS
### Study overview

Nine hundred patients aged 70 or over in EDs or acute general medical wards will be recruited in three sites (Edinburgh, Bradford and Sheffield). Study recruitment commenced on 19 October 2015. Recruitment was planned to be completed in December 2016, with final follow-up data collection and locking of the database in March 2017. The assessments are: (a) a reference standard delirium assessment lasting up to 20 min and (b) either the 4AT or the CAM lasting up to 10 min. The reference standard and 4AT or CAM assessments

will take place within a maximum of 2 hours of each other, with a target interval of 15 min. The results of the reference standard assessment were recorded in the case notes and communicated to the clinical team after the index assessments had been completed and recorded. The team will invite an appropriate informant to complete a questionnaire on participant's preadmission cognitive function. This will be completed within 4 weeks of the patient being recruited to the study assuming an appropriate individual is available.

At 12 weeks, the team will also administer a 10 min resource-use questionnaire (face to face in hospitalised patients or by telephone when possible) and will access each recruited patient's medical records at 12 weeks to ascertain a set of key clinical outcomes, including length of stay, institutionalisation and mortality, as well as to derive further information on resource utilisation. The study flow chart is shown in figure 1. The study has been registered: international standard randomised controlled trial number (ISRCTN) 53388093. UK Clinical Research Network ID: 19 502.

## Inclusion criteria
► Aged 70 or over.

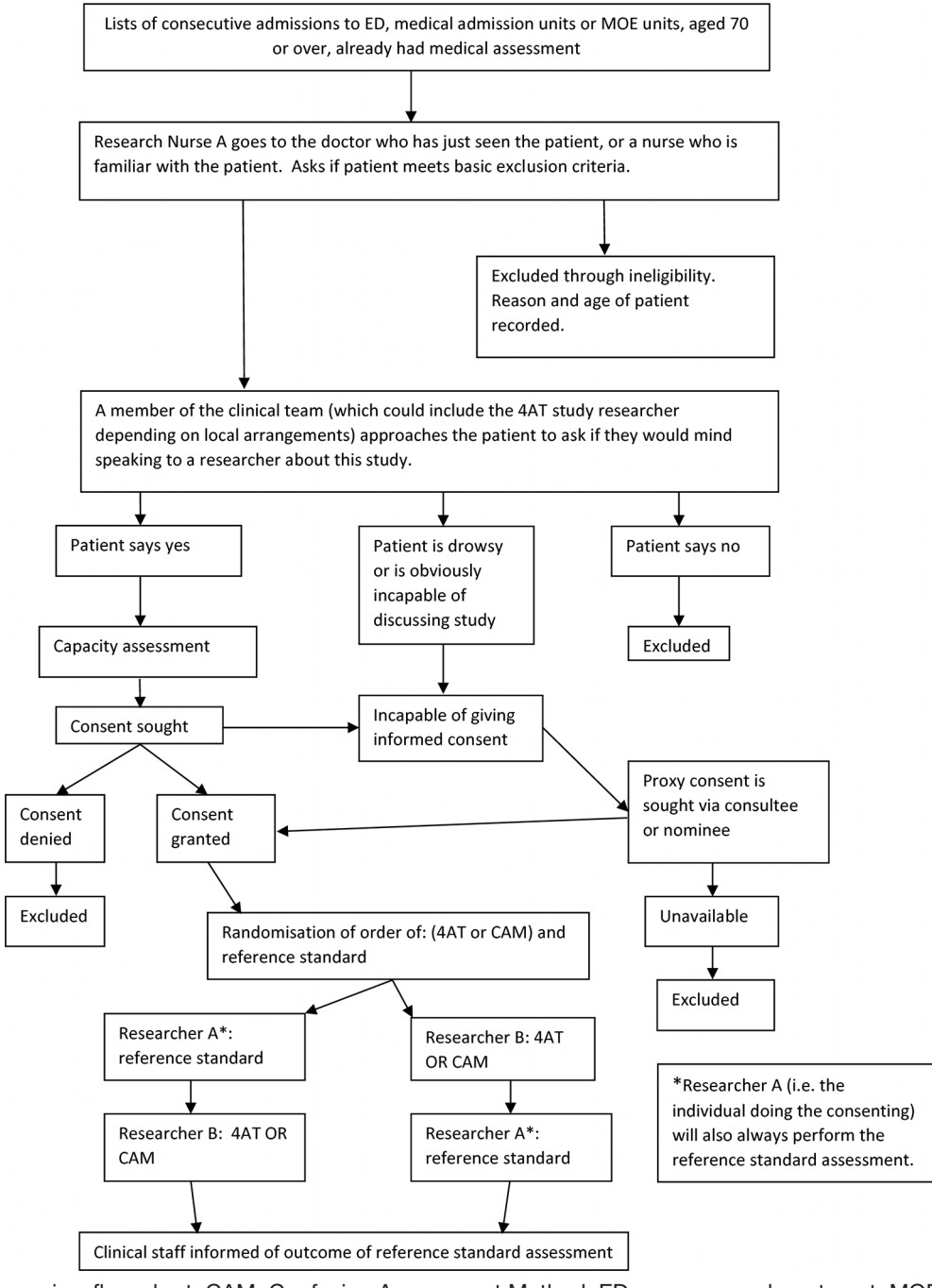

**Figure 1** Study overview flow chart. CAM, Confusion Assessment Method; ED, emergency department; MOE, Medicine of the Elderly; 4AT, 4A's Test.

► Acutely admitted to the ED (within 12 hours of attending) or acute general medical and geriatrics units (within 96 hours of admission to the ward). For ED patients, we will only recruit from those patients who were brought in by ambulance as an emergency or through their general practitioner.

## Exclusion criteria
► Acute life-threatening illness requiring time-critical intervention, for example, ST-elevation myocardial infarction, septic shock and severe pulmonary oedema.
► Coma ('Unresponsive' on the AVPU scale[43]).
► Unable to communicate in English or severe dysphasia.

## Identification of participants
The participant screening strategy in the initial protocol stated that patients will be recruited between 08.00 and 22.00. A list of potentially eligible patients will be generated in batches at the start of each recruitment period and initial eligibility screening will be carried out by clinical staff (including clinical research nurses embedded in the clinical team). Then, in alphabetical order, in each batch, consent/agreement from patient (or proxy/consultee) will be sought by a study researcher. Numbers of those (a) initially potentially eligible, (b) screened as non-eligible by clinical staff and (c) declining to take part will be recorded.

This recruitment strategy was modified in the last 5 months of recruitment because preliminary analyses suggested that patients at lower risk of delirium (ie, those not requiring capacity assessment) were more likely to be recruited than those with impaired capacity. Thus, to allow for some oversampling of patients at higher risk of delirium, a pragmatic approach was adopted. From the batches of patients identified as in the original strategy, patients considered at higher risk of delirium on clinical grounds (eg, older age, likely to be admitted, higher degree of ongoing acute and chronic illnesses) were approached first, rather than by alphabetical order.

## Assessing capacity and obtaining informed consent
Informed consent will be sought by a trained researcher using a combined informal capacity assessment/consent process.[44] Both verbal and written information will be provided about the study, using a style and format suitable for the participant group (ie, for varying levels of capacity). The researcher will ask the potential participant to recount the study which will be used, with the treating team views, to assess capacity to consent. For participants judged to have capacity, consent will be sought for:
a. conducting assessments as specified in the study information sheets;
b. accessing health records for information relevant to outcomes and health service use and
c. recording these data in secure study databases.

It will be made clear to participants that they are under no obligation to take part, their usual care will not be affected by their decision and they can withdraw consent without giving a reason. Once participants are enrolled in the study, they will be given a sheet with contact details for the research team and instructions on what to do if they wish to withdraw consent or require further information.

## Lack of capacity to consent
It is essential that this study recruits patients which reflect the target clinical population. This means that we must recruit patients with delirium in the same proportion as in the clinical population. We will seek consent/agreement from legal proxies, consultees or other legal representatives. Where the potential participant is deemed to lack capacity to consent, recruitment will proceed under the provisions of the Mental Capacity Act, 2005 in England or Adults with Incapacity (Scotland) Act, 2000. The clinical team will be asked to identify an appropriate personal or nominated consultee, guardian, welfare attorney or nearest relative.

Because of differing legal requirements in Scotland and England, the details of the processes in each nation are given in online Supplementary appendix 1.

## Interventions to be measured
Assessments will be carried out by researchers fully trained in background information on delirium, the features of delirium and each rating scale. Training is carried out using written, video and bedside training until competence in all aspects of the assessments is achieved.

### The 4 A's Test
The 4AT (see www.the4AT.com) comprises four items. Item 1 concerns an observational assessment of level of alertness. The next two items are brief cognitive tests: the Abbreviated Mental Test–4 (AMT4) which asks the patient to state their age, their date of birth, the current year and the place they are in, and attention testing with Months Backwards, in which the patient is asked to state the months of year in reverse order, starting with December. Only items 1–3 are done at the bedside, and the typical duration is under 2 min. Item 4 concerns acute change in mental status, a core diagnostic feature of delirium; this information is obtained from the case notes or the general practitioner (GP) letter or from an informant.

### Short CAM
The CAM is a diagnostic algorithm in which the tester rates the following four features as positive or negative: (1) Acute Change and Fluctuating Course; (2) Inattention; (3) Disorganised Thinking and (4) Altered Level of Consciousness. The CAM scoring process requires that Features 1 and 2 are both positive; if they are positive, then Features 3 and 4 are assessed and if one of Features 3 or 4 is positive, then the whole CAM is positive. The tester scores the features by a combination of interview with the patient, cognitive testing (the CAM requires that a cognitive test is performed before the features are scored), examining the case notes and seeking informant history

if required. Note that the questionnaires used to assess cognition are not specified by the CAM manual, though some suggested tests are provided. Feature 1 is assessed by the same process as Item 4 in the 4AT. Feature 2 is assessed by the tester giving a positive or negative rating to the question, 'Did the patient have difficulty focusing attention, for example, being easily distractible or having difficulty keeping track of what was being said?' Feature 3 is assessed by the tester giving a positive or negative rating to the question, 'Was the patient's thinking disorganised or incoherent, such as rambling or irrelevant conversation, unclear or illogical flow of ideas or unpredictable switching from subject to subject?' Feature 4 is similar to Item 1 in the 4AT. In this study, for the pre-CAM cognitive assessment, we will use a set of questions covering the cognitive domains represented in the suggested tests in the CAM manual, including Days of the Week Backwards, counting from 20 down to 1, orientation questions, three-word recall and clockdrawing, as well as simple orientation questions. All of these questions are used in routine clinical practice at the bedside.

### Reference standard assessment

Reference standard assessment: this will be centred on the Delirium Rating Scale-Revised-98 (DRS-R98),[45] requiring inspection of case notes, speaking to staff who know the patient or speaking to the patient's relatives or others who know them (with patient consent). As per the instruction manual, the DRS-R98 will be supplemented by short neuropsychological tests of attention and other domains, including Digit Span,[27] the Observational Scale for Level of Arousal,[46] the Richmond Agitation Sedation Scale[47] and the DelApp objective attentional assessment.[48] We will also perform simple orientation questions and record any formal prior diagnosis of dementia and IQCODE[49] scores. The DRS-R98 and supporting tests will be used to inform a binary ascertainment of delirium based on DSM-IV criteria. The final DSM-IV ascertainment of delirium will be based on a standardised process with final verification by the chief investigator, blind to the 4AT or CAM results. The panel of supporting tests, and the way the data are coded will be designed such that the performance of the 4AT can also be evaluated against the DSM-5 criteria.[2] The reference standard assessment will take approximately 15–20 min.

### Ordering of assessments

All patients will undergo a reference standard assessment for delirium by the researcher who conducted the capacity assessment and consenting process. A different researcher will also ask each patient to undergo either the 4AT or the CAM. The reason that the researcher doing the capacity assessment and consenting process must also do the reference standard assessment is that the capacity and consenting process provides information to the tester over and above the normal 4AT or CAM testing process. This is not a concern for the reference standard assessment, which is aimed at providing a thorough assessment

so as to optimise diagnostic accuracy. The order of these two assessments ((4AT or CAM assessment) and reference standard assessment) will be randomly allocated immediately after consenting, as will the assignment to either the 4AT or the CAM. Each patient will receive the reference standard assessment by the same researcher who did the capacity and consenting process. The 4AT or CAM will be performed by a different researcher. When possible, the IQCODE will then be administered to a person who knows the patient well (within 4 weeks of the patient joining the study).

### Randomisation procedure

The allocation sequence will be created using computer-generated random numbers. Participants will be randomised in a 1:1 ratio to be assessed using the 4AT or CAM experimental assessment. The order in which they receive the reference standard and experimental assessment will also be randomised in a 1:1 ratio. Randomisation will be stratified by study site with block allocation. The randomised allocations will be concealed until they are assigned, as the randomisation system will be web based and requires a personal login and password. Once randomisation has been performed, neither the researchers nor the participant will be blinded to the allocation as both will be aware of the assessments conducted and the order in which they are performed.

### Outcome measurements (what, when, how)

Note that the outcome measurements for the primary study (the reference standard) are performed at almost the same time as the 4AT and CAM. The only subsequent data collection is capturing clinical outcomes at 12 weeks. This will be achieved through searching electronic patient records, telephone calls with participants or face-to-face interviews if still in hospital. The information gleaned at the 12-week point will at times be generalised due to participant recall or availability of full electronic records.

(1) Primary outcome measure:

Diagnostic accuracy of the 4AT versus the reference standard delirium diagnosis

1. Secondary outcome measures:
2. 4AT versus CAM in relation to reference standard delirium diagnosis.
3. Performance of 4AT cognitive test items (AMT4 and Months Backwards) in detecting longer term cognitive impairment as detected by the IQCODE.
4. 4AT total scores as a predictor of the following clinical outcomes as determined at 12 weeks post-test: length of stay, falls, institutionalisation (as assessed by proportion of patients newly admitted to care homes or awaiting care homes at that time) and mortality.
5. Performance of individual items of the 4AT in relation to reference standard delirium diagnosis.
6. We will assess the 4AT total score as a measure of delirium severity.

7. The primary output from the health economic analysis will be a comparison of the service delivery costs associated with the diagnostic accuracy of alternative (4AT vs CAM vs reference standard) triage tools for delirium.

## Coding and recording assessments

The experimental assessments of delirium will be the 4AT and the CAM. The 4AT has a total possible score of 12: items (1) and (4) can score 0 or 4 and items (2) and (3) can score 0, 1 or 2.[12] 4AT data will be used for the primary objective as a binary outcome, with 0–3 scores giving a 'no delirium' classification, and 4–12 scores giving a 'delirium' classification; for the secondary objectives, continuous scoring, from 0 to 12, will be studied as a possible severity indicator, and scores of 1–3 (indicating cognitive impairment but not delirium) can be studied against other assessments of chronic cognitive impairment. The CAM will be scored as delirium present or absent according to the algorithm. The 4AT, CAM scoring and reference standard scoring will be recorded on a paper Case Report Form.

Patient resource use will be derived from medical records, including the 'TrakCare' (InterSystems Corporation, Cambridge, Massachusetts, USA) electronic patient record system, where available, as well as via patient or carer self-report. The self-report resource-use questionnaire will include questions regarding inpatient health and social care utilisation with a maximum recall period of 16 weeks. The self-report resource-use questionnaire will be developed specifically for the study for use by patient or proxy respondent using guidance from the Database of Instruments for Resource Use Measurement.[50] Administration of the questionnaire will be conducted at 12 weeks by one of the researchers in the study team, face to face where patients are still hospitalised or via telephone. Data from the questionnaire will be recorded on a paper Case Report Form.

The data on all the Case Report Forms will be transcribed into a secure database by the researchers or a suitably qualified member of the research team. This will be conducted using Edinburgh Clinical Trials Unit Standard Operating Procedures. Quality checking will be performed in 10% of Case Report Forms.

## Sample size calculation

Four hundred and fifty patients will be randomised to assessment by 4AT and 450 to CAM. We will recruit sufficient patients to account for attrition, though we do not expect significant attrition because the recruitment, consenting and assessment process takes place over a small number of hours in a single episode. Of the 450 patients within each assessment arm, 15% (67) would be expected to have delirium. The specificity of the triage tool would be estimated based on the 85% (383) without delirium, while the sensitivity would be estimated from the 67 with delirium. Based on the analysis using the normal approximation to the binomial distribution, the

### Table 1 Precision of specificity, sensitivity estimation

| Parameter | True level of parameter | 95% CI width |
|---|---|---|
| Specificity | 0.5 | ±0.050 |
| Specificity | 0.7 | ±0.046 |
| Specificity | 0.9 | ±0.030 |
| Sensitivity | 0.5 | ±0.120 |
| Sensitivity | 0.7 | ±0.110 |
| Sensitivity | 0.9 | ±0.072 |

two-sided 95% CI widths for the specificity and sensitivity would be as shown in table 1 for a range of levels of diagnostic test performance.

It will therefore be possible to estimate the specificity precisely and the sensitivity with moderate precision. The precision in estimating negative predictive value would be expected to be similar to that for specificity; for positive predictive value, it would be expected to be similar to that for sensitivity. For the secondary objective of comparing 4AT and CAM, based on analysis by continuity corrected $\chi^2$ test, we have 83% power to detect a difference in specificity of 0.1, assuming a null hypothesis of specificity=0.70 for both tests and a two-sided 5% significance level. The corresponding difference detectable for sensitivity (null hypothesis sensitivity=0.7) would be 0.224 with 80% power.

## Data analysis plan

The detailed statistical analysis plan (SAP) will be agreed prior to database lock and will be prepared by individuals blinded to the randomised allocations.

### Primary objective

(a) 4AT versus reference standard: the diagnostic accuracy of 4AT versus the reference standard will be assessed using positive and negative predictive values, sensitivity and specificity. The exact binomial 95% CI will be reported for each measure. A ROC curve will be constructed to verify that the proposed cut point of greater than 3 on the 4AT score is appropriate. The area under the ROC curve and its 95% CI will be reported.

### Secondary objectives

a. 4AT versus CAM: differences in each of sensitivity, specificity, positive and negative predictive values between 4AT and CAM will be tested by Fisher's exact test and quantified by the difference in the two proportions (4AT-CAM) and its 95% CI. To aid comparison of 4AT and CAM, the overall performance of each will also be summarised using Youden's Index (sensitivity minus false positive rate) and the OR of sensitivity to specificity.

b. Performance of the 4AT cognitive screening items: Is the 4AT an adequately sensitive tool for detecting general cognitive impairment as judged against a documented history of dementia and/or the

IQCODE? This objective will be addressed using the same methods as for the primary objective.

c. 4AT versus clinical outcomes: as assessment of criterion validity, we will assess the performance of the 4AT in predicting length of stay, institutionalisation and mortality up to 12 weeks. Descriptive statistics of clinical outcomes will be presented for the groups with and without 4AT scores above the cut point of 3. The relationship between 4AT and each of mortality and institutionalisation will be analysed via logistic regression modelling; Kaplan-Meier curves and the Cox proportional hazards model will be used to assess 4AT as a predictor of hospital length of stay. The logistic regression and Cox models will adjust for age, gender and presence of dementia.

d. Individual items: we will conduct analyses examining performance of individual items of the 4AT, for example, is altered level of alertness alone a good predictor of delirium diagnosis? (methods as per primary objective);

e. Delirium severity: we will assess the 4AT total score as a measure of delirium severity by calculating the Spearman correlation between 4AT and DRS-R98 scores and its 95% CI.

Full details of the proposed statistical analyses for the primary objective and secondary objectives (a)–(e) will be documented in a SAP which will include details of methods for calculating derived variables, any sensitivity and subgroup analyses and approaches to testing the assumptions in the statistical analyses. The SAP will outline the plan for validation of the statistical analysis.

Individuals with missing data for the reference diagnostic test will be removed from formal statistical analysis. Where any items of the CAM or 4AT were not able to be assessed, an overall delirium diagnosis will still be derived where possible based on the items which have been recorded. There will be no other imputation of missing delirium diagnoses.

(f) Delivery costs of the triage tools: we will estimate the delivery costs and subsequent resource consequences associated with the triage tools as a function of sensitivity and specificity from the perspective of the UK National Health Service (NHS). Potential resource consequences may include additional diagnostic procedures (eg, more detailed cognitive screening and brain imaging), altered management as well as readmissions. Healthcare resource use will be derived from medical records, including the 'TrakCare' (InterSystems Corporation, Cambridge, Massachusetts, USA) electronic patient record system, where available, as well as via patient or carer self-report. Monetary values will be attached to resource use, training and labour costs as well as the indirect costs of delivering each diagnostic tool using standard NHS pay and price estimates. Generalised linear models will be used to analyse 12 week cumulative costs which will inform longer term resource consequences within a decision analytic model.

### Study oversight

Study oversight is through the trial steering committee, which will meet every 4 months during the study. The trial steering committee comprises two independent lay representatives, three independent experts (one of whom is the Chair of the committee), the principal investigator (PI), the study statistician and representatives from the Edinburgh Clinical Trials Unit.

### Data protection

Data will be collected and handled in line with sponsor and Edinburgh Clinical Trials Unit Standard Operating Procedures and NHS Trust policies. All electronic data will be link anonymised.

### DISCUSSION

This study was designed to validate the 4AT against a reference standard assessment, as well as compare it with another commonly used test for assessment of delirium. Since the initial study design, the 4AT has been widely adopted nationally and internationally. The 4AT has been incorporated into routine practice in multiple international centres, both in paper and electronic format, with many centres reporting 10 000 uses of the tool. The website www.the4AT.com has had an increasing degree of traffic, and the 4AT has been translated into multiple languages. The 4AT is also included in several national guidelines and position statements internationally as a recommended tool and it has been validated in other studies[38–42] but it is still essential that it is further tested in a large study. It is also essential to consider how well it identifies other types of cognitive impairment, relates to future outcomes and its health economic impact.

In the initial design and implementation of the study, the main challenging aspects have been: (1) considering both Scottish and English legal and ethical framework to ensure that patients without capacity are included. Ethical approval for inclusion of these patients was granted though recruitment of this patient group proved difficult from the outset for several reasons. First, the narrow boundaries for the screening and identification strategy. This was addressed in a subsequent protocol amendment to aim for oversampling of patients at risk of developing delirium. Second, the availability of an appropriate individual to provide consent on behalf of the participant (ie, a personal or nominated consultee, guardian, welfare attorney or nearest relative). Third, a reluctance to consent due to perceived burden on participant. Persuading relatives of the value, importance and necessity of research even in clinically unwell patients demands a particular skillset from researchers and involved perseverance and excellent communication in order to achieve recruitment targets.

(2) Recruitment and training of staff, with some staff moving to different posts, and new staff being recruited and requiring training; in each case, detailed training

supported by reading materials and practice sessions was provided.

We also acknowledge that it is possible that researcher bias may influence how the different index assessments (4AT or CAM) were scored. We also acknowledge that given the fluctuating nature of delirium, the gap between assessments potentially reaching 2 hours means that assessments could have different findings. We will conduct sensitivity analyses to analyse the impact of variations in the time gap between assessments.

## CONCLUSION

The 4AT study aims to assess the validity of this rapid delirium screening tool that can be administered by a range of professional- level healthcare staff to patients with sensory or functional impairments in a busy clinical environment, which also incorporates general cognitive assessment. We will also assess the later functional outcomes of people with and without delirium and the health economic implications. The overall aim is to improve detection, and therefore management and outcomes, of this important and devastating condition.

**Author affiliations**
[1]Geriatric Medicine, University of Edinburgh, Edinburgh, UK
[2]Old Age Psychiatry, University of East Anglia, Norfolk, UK
[3]Health and Social Care, Institute of Health Sciences, University of Leeds, Leeds, UK
[4]Psychiatry, University of York, York, Hull York Medical School, York and Bradford District Care NHS Foundation Trust, Bradford, UK
[5]Emergency Medicine, University of Sheffield, Sheffield, UK
[6]Elderly Care and Rehabilitation, University of Leeds, Leeds, UK
[7]Cardiovascular Sciences and Geriatric Medicine, University of Edinburgh, Edinburgh, UK
[8]Professor of Emergency Medicine, Department of Emergency Medicine, Emergency Medicine Research Group (EMERGE), NHS Lothian, Edinburgh, UK
[9]Nuffield Department of Population Health, Health Economics Research Centre, University of Oxford, Oxford, UK
[10]Old Age Liaison Psychiatry, NHS Lothian, Edinburgh, UK
[11]Health and Social Care, Edinburgh Napier University, Edinburgh, UK
[12]Emergency Medicine Research Group Edinburgh (EMERGE), NHS Lothian, Edinburgh, UK
[13]Edinburgh Clinical Trials Unit, University of Edinburgh, Edinburgh, UK
[14]Usher Institute of Population Health Sciences and Informatics, University of Edinburgh, Edinburgh, UK
[15]Geriatric Medicine, University of Edinburgh, Edinburgh, UK

**Acknowledgements** Thanks to the trial steering committee.

**Contributors** All authors were involved in the design of the protocol for the study. AMJM initiated the conception of the study. AM, JS and PLB were involved in protocol design and acquisition of data; CJW initiated the design of the statistical analysis plan. JS initiated the design of the economic analysis. SDS and AMJM drafted the paper and all authors revised it critically for important intellectual content. All authors have approved the final approval of the version to be published and agree to be accountable for all aspects of the work. AMJM is the guarantor.

**Funding** This work was funded by National Institute of Health Research Technology Assessment Programme (NIHR DTA) grant number 11/143/01 (PI: AMJM; CO-I: SG, JS, AA, MG, SDS, TR, NS, CF, JH, CJW, AG). JB and CJW were supported in this work by NHS Lothian via the Edinburgh Clinical Trials Unit. AA has received personal fees from Abbott Diagnostics, outside the submitted work. SG is funded by NIHR to undertake his role as NIHR HTA Clinical Evaluation and Trials Board Chair. SDS is funded by NRS (NHS Research Scotland) to undertake her role as NRS Ageing Specialty Group Lead (Scotland). AM and SDS are members of the University of Edinburgh Centre for Cognitive Ageing and Cognitive Epidemiology (funded by the BBSRC and MRC as part of the LLHW [MR/K026992/1]).

**Competing interests** No competing interests.

**Ethics approval** This study was granted ethical approval in Scotland(Scotland A NHS research ethics committeeREC 15/SS/0071) and England (Yorkshire and The Humber–BradfordLeeds NHS research ethics committeeREC 15/YH/0317).

**Provenance and peer review** Not commissioned; externally peer reviewed.

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
