## [Reviewer comments · BMJ Open]

ARTICLE DETAILS

TITLE (PROVISIONAL)	Protocol for validation of the 4AT, a rapid screening tool for delirium: a multicentre prospective diagnostic test accuracy study
AUTHORS	Shenkin, Susan Fox, Chris Godfrey, Mary Siddiqi, Najma Goodacre, Steve Young, John Anand, Atul Gray, Alasdair Smith, Joel Ryan, Tracy Hanley, Janet MacRaid, Allan Steven, Jill Black, Polly Boyd, Julia Weir, Christopher MacLulich, Alasdair

VERSION 1 - REVIEW

REVIEWER	Alessandro Morandi
REVIEW RETURNED	16-Jan-2017

GENERAL COMMENTS	This is the protocol of a multicenter ongoing study aiming primarily to validate the 4AT against a reference standard. Secondary aims include include (a) comparing the 4AT with another widely used test (the Confusion Assessment Method (CAM)); b) determining if the 4AT is sensitive for general cognitive impairment; c) assessing if 4AT scores predict outcomes; including d) a health economic analysis. The manuscript is well written and the study is well designed. The study will provide important information on the clinical and research use of the 4-AT.
---

REVIEWER	Peter G Lawlor
REVIEW RETURNED	17-Feb-2017

GENERAL COMMENTS	This is a very worthwhile study, especially in light of current and future population demographic changes that will result in an increase in the overall of number of patients presenting with delirium and dementia. There is already a compelling need to have access to a validated delirium and cognitive screening tool with good psychometric properties, yet have sufficient brevity to make its use acceptable across a variety of clinical settings.
---

	This validation study is ambitious on many fronts: the tool itself incorporates a novel hybrid function in the dual assessment of delirium and cognitive function; the study's projected large sample size of 900 offers adequate power, pending the limits of attrition and missing data, which could prove substantial; finally, its aim to assess the performance of the 4AT in one of the busiest and chaotic sites of healthcare, the Emergency Department will be a true test of its clinical utility. Basically, a good 4AT performance in the Emergency Department would bode well for other sites. The protocol is clearly written albeit submitted late in the study, which must now be almost closed. My comments may have some relevance to the interpretation of the study findings. One of my biggest concerns relates to the utility of the 4AT tool in the sicker or more frail patient. Patient recruitment in these types of studies tends to naturally favour the healthier patient. The authors have addressed this point in their manuscript; they report making adjustments to include some of the sicker patients. Although the randomised study design, including allocation arm and sequence of testing, helps to minimise bias, there remains some concern regarding the potential researcher bias in unwittingly favouring one tool over the other. Also, the 2-hour maximum for the interval between reference standard assessment and the 4AT or CAM is certainly large enough to allow score differences due to the natural fluctuation of delirium rather than the test tool properties. A sensitivity analysis could be conducted to examine the outcomes in relation to the interval duration, assuming that the interval has been accurately recorded. I appreciate that the target of 15 minutes may be difficult to achieve. The monitoring of this study is clearly described appears to have been well conducted. Also, the appropriate ethical approvals have been obtained for this study. I think that the authors should state clearly if the care team were made aware of the scores or assessments once they are completed. It would seem ethically plausible and desirable to do so; such information could impact some of the later outcomes. Overall, this is scientifically a very well designed study. I eagerly look forward to some informative reports in due course.
--	---

VERSION 1 – AUTHOR RESPONSE

Reviewer: 1

Reviewer Name: Alessandro Morandi

Institution and Country: Department of Rehabilitation and Aged Care Unit, Ancelle Hospital, Cremona, Italy

Competing Interests: None declared

This is the protocol of a multicenter ongoing study aiming primarily to validate the 4AT against a reference standard. Secondary aims include include (a) comparing the 4AT with another widely used test (the Confusion Assessment Method (CAM)); b) determining if the 4AT is sensitive for general cognitive impairment; c) assessing if 4AT scores predict outcomes; including d) a health economic analysis. The manuscript is well written and the study is well designed. The study will provide important information on the clinical and research use of the 4-AT.

Thank you

Reviewer: 2

Reviewer Name: Peter G Lawlor

Institution and Country: University of Ottawa, Canada Competing Interests: None declared

This is a very worthwhile study, especially in light of current and future population demographic changes that will result in an increase in the overall of number of patients presenting with delirium and dementia. There is already a compelling need to have access to a validated delirium and cognitive screening tool with good psychometric properties, yet have sufficient brevity to make its use acceptable across a variety of clinical settings.

This validation study is ambitious on many fronts: the tool itself incorporates a novel hybrid function in the dual assessment of delirium and cognitive function; the study's projected large sample size of 900 offers adequate power, pending the limits of attrition and missing data, which could prove substantial; finally, its aim to assess the performance of the 4AT in one of the busiest and chaotic sites of healthcare, the Emergency Department will be a true test of its clinical utility. Basically, a good 4AT performance in the Emergency Department would bode well for other sites.

The protocol is clearly written albeit submitted late in the study, which must now be almost closed. My comments may have some relevance to the interpretation of the study findings. One of my biggest concerns relates to the utility of the 4AT tool in the sicker or more frail patient. Patient recruitment in these types of studies tends to naturally favour the healthier patient. The authors have addressed this point in their manuscript; they report making adjustments to include some of the sicker patients. Although the randomised study design, including allocation arm and sequence of testing, helps to minimise bias, there remains some concern regarding the potential researcher bias in unwittingly favouring one tool over the other. Also, the 2-hour maximum for the interval between reference standard assessment and the 4AT or CAM is certainly large enough to allow score differences due to the natural fluctuation of delirium rather than the test tool properties. A sensitivity analysis could be conducted to examine the outcomes in relation to the interval duration, assuming that the interval has been accurately recorded. I appreciate that the target of 15 minutes may be difficult to achieve.

As the reviewer states, we previously adjusted the protocol to help ensure that we recruited a more representative sample, and this is already stated in the manuscript ('Identification of participants' section).

We have added an acknowledgement that researcher bias could be a factor in the results, and also an acknowledgement that fluctuation might affect the results, along with a statement that we will conduct sensitivity analyses to examine the potential effects of variations in the time gap between assessments. (Discussion):

"We also acknowledge that it is possible that researcher bias may influence how the different index assessments (4AT or CAM) were scored. We also acknowledge that given the fluctuating nature of delirium, the gap between assessments potentially reaching two hours means that assessments could have different findings. We will conduct sensitivity analyses to analyse the impact of variations in the time gap between assessments"

The monitoring of this study is clearly described appears to have been well conducted. Also, the appropriate ethical approvals have been obtained for this study. I think that the authors should state clearly if the care team were made aware of the scores or assessments once they are completed. It would seem ethically plausible and desirable to do so; such information could impact some of the later outcomes.

We have added a statement to this effect in the Methods and Analysis/Study Overview Section (paragraph 1):

"The results of the reference standard assessment were recorded in the casenotes and communicated to the clinical team after the index assessments had been completed and recorded."

Overall, this is scientifically a very well designed study. I eagerly look forward to some informative reports in due course.